# DeiT and Image Deep Learning-Driven Correction of Particle Size Effect: A Novel Approach to Improving NIRS-XRF Coal Quality Analysis Accuracy

**DOI:** 10.3390/s25030928

**Published:** 2025-02-04

**Authors:** Jiaxin Yin, Ruonan Liu, Wangbao Yin, Suotang Jia, Lei Zhang

**Affiliations:** 1State Key Laboratory of Quantum Optics and Quantum Optics Devices, Institute of Laser Spectroscopy, Shanxi University, Taiyuan 030006, China; 202222618045@email.sxu.edu.cn (J.Y.); 202322618022@email.sxu.edu.cn (R.L.); tjia@sxu.edu.cn (S.J.); 2Collaborative Innovation Center of Extreme Optics, Shanxi University, Taiyuan 030006, China

**Keywords:** coal quality analysis, near-infrared spectroscopy (NIRS), X-ray fluorescence (XRF), particle size effect, image segment, data-efficient image transformer (DeiT)

## Abstract

Coal, as a vital global energy resource, directly impacts the efficiency of power generation and environmental protection. Thus, rapid and accurate coal quality analysis is essential to promote its clean and efficient utilization. However, combined near-infrared spectroscopy and X-ray fluorescence (NIRS-XRF) spectroscopy often suffer from the particle size effect of coal samples, resulting in unstable and inaccurate analytical outcomes. This study introduces a novel correction method combining the Segment Anything Model (SAM) for precise particle segmentation and Data-Efficient Image Transformers (DeiTs) to analyze the relationship between particle size and ash measurement errors. Microscopic images of coal samples are processed with SAM to generate binary mask images reflecting particle size characteristics. These masks are analyzed using the DeiT model with transfer learning, building an effective correction model. Experiments show a 22% reduction in standard deviation (SD) and root mean square error (RMSE), significantly enhancing ash prediction accuracy and consistency. This approach integrates cutting-edge image processing and deep learning, effectively reducing submillimeter particle size effects, improving model adaptability, and enhancing measurement reliability. It also holds potential for broader applications in analyzing complex samples, advancing automation and efficiency in online analytical systems, and driving innovation across industries.

## 1. Introduction

Coal, as a critical global energy resource, directly impacts the efficiency of power generation, coal preparation, and coal chemical industries, as well as their environmental effects [1,2]. Therefore, rapid and accurate coal quality analysis plays a crucial role in promoting the clean and efficient utilization of coal [3]. Key indicators of coal quality include ash, volatile matter, calorific value, and sulfur content, which determine the combustion performance and environmental characteristics of coal [4].

Traditional coal quality analysis methods include LIBS [5,6], XRF [7], NIRS [8] and Dual-Energy X-ray Analysis (DEXA) [9,10]. These methods, with their high sensitivity, rapid response, and non-destructive characteristics, offer certain advantages in terms of accuracy. However, they often face challenges such as complex sample preparation, lengthy analysis times, and susceptibility to human interference. To address these issues, NIRS-XRF combined spectroscopy [11] has emerged as an analytical technique that integrates the strengths of near-infrared spectroscopy (NIRS) and X-ray fluorescence spectroscopy (XRF). It provides higher accuracy and reliability, particularly in overcoming the limitations of single techniques [12].

NIRS excites molecular vibrations in the sample using specific wavelengths of light, generating unique absorption spectra that precisely reflect the organic components of the sample [13]. Meanwhile, XRF employs high-energy X-rays to excite atoms in the sample, producing characteristic fluorescence spectra that efficiently and reliably measure inorganic elements [14]. The combination of these two techniques enables highly stable and comprehensive detection of coal components, facilitating the rapid and accurate analysis of key coal quality indicators. For example, Gao et al. [15] developed a fast calorific value analyzer for coal using NIRS-XRF technology, employing a partial least squares regression (PLSR) algorithm with an overall-segmented model. They achieved a standard deviation of 0.09 MJ/kg when measuring the calorific value of four coal products with a particle size of 0.2 mm. Additionally, they proposed a method for identifying coal types using random forests and applying corresponding PLSR sub-models to predict calorific value, effectively addressing the measurement challenges posed by varying coal types in complex applications like coking and coal washing industries. Similarly, Li et al. [16] proposed an automatic classification method for coking coal by combining NIRS-XRF fusion spectroscopy, principal component analysis (PCA), and t-distributed stochastic neighbor embedding (t-SNE) for dimensionality reduction. They classified the samples using support vector machines (SVM) and built regression models with PLSR. This approach significantly improved the prediction accuracy of ash, volatile matter, and sulfur content in coal samples with a particle size of 0.2 mm. The determination coefficient (R^2^) for ash reached 0.9987, with a root mean square error of prediction (RMSEP) of 0.31%.

In practical applications, grinding samples to a 0.2 mm particle size often causes blockages in grinders, leading to equipment downtime and increased maintenance costs, which in turn affect the overall operational efficiency of the system. In contrast, coal samples with a particle size of 1mm can be obtained directly through simple crushing processes, which are more convenient and significantly improve the efficiency of online coal detection. However, the NIRS-XRF analysis of 1mm coal samples is subject to interference from uneven surface particle distribution and particle size variations [17,18]. This is because a 1 mm particle size represents the D50 median diameter, meaning 50% of particles are smaller than or equal to this size. Larger particles may cause uneven light scattering and absorption [19]. To improve the accuracy and consistency of NIRS-XRF analysis for coal samples with a particle size of 1 mm or larger, effective correction of particle size effects has become a key issue that needs urgent resolution.

Conventional particle size correction methods, such as Multiplicative Scatter Correction (MSC) [20], Polynomial Multiplicative Scatter Correction (PMSC) [21], Standard Normal Variate Correction) [22], and Extended Multiplicative Scatter Correction (EMSC) [23], can partially reduce the impact of physical factors on spectral data and are widely used in industries such as agriculture [24] and pharmaceuticals [25]. However, these methods primarily rely on spectral data for correction and fail to account for the influence of spatial distribution and morphological characteristics of particles. In contrast, image segmentation technology can directly capture the spatial distribution and morphological information of sample particles, allowing for more precise correction of spectral changes caused by uneven particle size, thus providing more accurate correction results.

This study focuses on developing a particle size effect correction method based on machine vision and image deep learning, aiming to make NIRS-XRF prediction models insensitive to variations in particle size distribution. Specifically, the Segment Anything Model (SAM) [26] is used to achieve precise particle segmentation of microscopic images of coal samples, generating binary mask images that reflect particle size distribution characteristics. Then, Data-Efficient Image Transformers (DeiTs) [27] are employed to train these mask images and the associated ash measurement errors, establishing a particle size effect correction model for NIRS-XRF coal quality measurements. By validating the effectiveness of this method through experiments, we aim to provide an innovative solution to the particle size effect issue in spectral analysis.

## 2. Materials and Methods

### 2.1. Experiment

#### 2.1.1. Experiment Setup

The NIRS-XRF combined coal quality analysis experimental setup used for particle size effect correction comprises the following five modules as shown in Figure 1:

NIRS Module: A Fourier-transform infrared spectrometer (C15511-01, Hamamatsu Photonics, Hamamatsu, Japan) is employed, with a working wavelength range of 1100–2500 nm, a spectral resolution of 5.7 nm, a signal-to-noise ratio of 10,000:1, and a spectral repeatability of ±0.5 nm. The exposure time is set to 15 s. The light source is a halogen lamp (AvaLight-HAL-S Mini, Avantes, Apeldoorn, The Netherlands) with a working wavelength range of 360–2500 nm, a color temperature of 2700 K, and a service life exceeding 13,000 h.XRF Module: This energy-dispersive structure is equipped with an X-ray tube (VF-50J, Varex Imaging, Salt Lake City, UT, USA) with a maximum power of 50 W and a rhodium target cathode. The tube voltage and current are set to 14 kV and 0.4 mA, respectively. A silicon drift detector (VIAMP, KETEK, Munich, Germany) with a graphene window is included, featuring a peak integration time of 0.1 μs and an integration time of 30 s. To prevent low-energy X-ray fluorescence signals from being absorbed by air, a hydrogen generator (SFH-300, Shen Fen Analytical Instruments, Shanghai, China) supplies hydrogen with 99.99% purity at a flow rate of 150 mL/min to the measurement chamber.Microscopic Imaging Module: Located between the sample pool’s initial position and the NIRS analysis module, this module includes a CCD microscope (RY-602, Renyue Electronics, Shanghai, China) and a ring-shaped auxiliary light. The microscope features 1× optical magnification, ~30× electronic magnification, and a working distance of 120 mm, with a field of view of 18 mm × 10 mm. The light provides an illumination of 55,000 Lux with an optimized angle to ensure uniform illumination on the coal sample surface.Sample Transport Module: The core component is a one-dimensional motorized linear stage located beneath the NIRS and XRF modules, with an effective travel distance of 600 mm, a repeatability of 0.03 mm, and a moving speed of 20 mm/s. The sample loading cell mounted on this module measures 130 mm × 30 mm × 10 mm.Analysis and Control Module: This module consists of a computer (equipped with an NVIDIA GeForce RTX 4090 GPU) and a programmable logic controller (PLC, SIEMENS S7-1200), responsible for timing control and data processing during the measurement process. The experimental operation software, developed on the LabVIEW platform, provides a user-friendly interface, allowing users to input sample information, select measurement methods, and monitor and display the progress and results in real time.

During the experiments, microscope images, NIRS spectra, and XRF energy spectra were collected from a central square area on the sample loading cell of the sample transport module, as shown in Figure 2. The left side shows the sample loading cell captured by a standard camera, while the right side provides a detailed view of the central square area (9 mm × 9 mm, green square) in the measurement region, captured by a microscope and highlighted by the green square. This area was used for collecting microscope images, NIRS spectra, and XRF energy spectra. We carefully calibrated the movement distances between these modules to ensure alignment of the same area for data collection, using adhesive labels and X-ray film for precise alignment.

The experimental setup operates as follows: first, microscopic imaging (~5 s) is performed, followed sequentially by NIRS (~15 s) and XRF detection (~30 s). The entire process takes approximately 1 min. After completion, the sample pool returns to its initial position, and all data are saved and recorded. Special attention is required to align the near-infrared and X-ray beam spots to the same size and ensure consistent sample irradiation areas. This ensures that the microscopic images can be accurately cropped and correspond to the respective spectra. The experimental environment is maintained at 22–25 °C with a humidity of 40–50%.

#### 2.1.2. Samples

In this study, eight coal samples were collected from the coal preparation plant of Shanxi Sunshine Coking Group, all with a median particle size of 1 mm. These samples were classified as coking coal according to the Chinese National Standard GB/T5751-2009 Classification of Coals in China. The ash content standard values were determined following the Chinese National Standard GB/T212-2008 Methods for Chemical Analysis of Coal to ensure accuracy and authority.

The previously established ash prediction model based on 0.2 mm coal samples was used as the standard model. The eight 1 mm samples were used for particle size effect correction to ensure the standard ash prediction model maintained accurate prediction capabilities across samples with varying particle size distributions.

Each coal sample was placed in the sample pool, leveled, and tested 100 times, resulting in 100 microscopic images and 100 initial ash measurement values for each sample. For model training, 90 images were randomly selected from each sample, with the remaining 10 images used for testing. Thus, the dataset consisted of 720 training images and 80 testing images.

### 2.2. Construction of the Particle Size Effect Correction Model

The overall construction process of the particle size effect correction model is shown in Figure 3. For the microscopic images of the coal samples, systematic preprocessing, image segmentation, and post-processing are performed to generate binary mask images. The difference between the predicted ash values from the standard model and the actual ash values is then calculated. This forms a dataset comprising 224 × 224 pixels mask images and ash difference values, used for training and testing the correction model.

#### 2.2.1. Image Preprocessing and Post-Processing

The original images cover a large area, so they are first center-cropped to reduce the pixel size to 1000 × 1000. To minimize noise, Gaussian filtering is applied for image smoothing. Since the spectral acquisition region is circular, a circular mask is used to process the images, ensuring the size matches the mask’s outer boundary while maintaining a pixel size of 1000 × 1000.

For accurate coal particle segmentation, the SAM model is utilized to produce high-precision panoptic segmentation maps and extract the mask for each coal particle. The segmentation masks are sorted and filtered to remove abnormal sizes and discard smaller masks in overlapping regions.

The filtered segmentation masks are combined into a single mask image. Using this mask, valid segmented coal particle regions are extracted from the original image and resized to 224 × 224 pixels for subsequent model training and testing.

#### 2.2.2. Segment Anything Model

Image segmentation is a core task in computer vision [28], aiming to divide an image into multiple meaningful regions or objects for further analysis or processing. Traditional image segmentation methods, such as thresholding, edge detection [29], region growing [29], watershed algorithms [30] and normalized cuts [31], rely on handcrafted features and low-level image information. While effective for simple scenarios, these methods often fail when handling complex or irregularly shaped objects. With advancements in deep learning, methods such as U-Net [32] and Mask R-CNN [33] have significantly improved segmentation capabilities through large-scale data training and end-to-end learning. However, these methods typically require extensive labeled data and task-specific architecture design, limiting their generalizability and applicability across tasks.

Against this backdrop, the Segment Anything Model (SAM) emerges as an innovative image segmentation method. Trained on vast amounts of data, SAM captures diverse visual concepts and supports cross-domain transfer learning, greatly enhancing its generality for segmentation tasks in various applications [26]. Its key advantage lies in its ability to perform efficient and precise segmentation even in the absence of abundant labeled data, automatically adapting to different scenarios. This capability is particularly valuable in complex, dynamic environments. SAM has been widely applied in areas such as medical image analysis [34], remote sensing image processing [35], object detection in autonomous driving [36], industrial inspection [37] and agricultural pest identification [38,39]. Its exceptional flexibility and efficiency make it particularly suitable for tasks involving complex structures or diverse environments.

The SAM model combines the strengths of CNNs and transformers to form an efficient image segmentation method. CNNs excel at extracting local features and recognizing spatial structural information within images, making them particularly suitable for tasks with prominent local patterns. At the same time, transformers, through their self-attention mechanism, capture global information and exploit long-range dependencies in images. This combination allows SAM to extract fine details while also understanding the global context of an image, achieving more precise and robust segmentation. In particular, when processing the panoptic segmentation of coal particle images, SAM effectively addresses differences in particle morphology and the complexity of image backgrounds, providing more accurate segmentation results.

Figure 4 illustrates the basic structure of the SAM model [26], which consists of three main components: an Image Encoder, a Prompt Encoder, and a Mask Decoder. First, the Image Encoder (based on the Vision Transformer) extracts features from the input image, generating an embedded representation of the image (Image Embedding). Next, the Prompt Encoder converts user-provided prompts (such as points, bounding boxes, or text) into embeddings. These embeddings are combined with the image embeddings and input into the Mask Decoder. Finally, the Mask Decoder integrates the image features and prompt information to generate the final segmentation mask. This overall structure effectively combines global image features with the guidance of prompt information, enabling the SAM model to flexibly adapt to various segmentation tasks.

In this study, SAM is applied to the task of panoptic segmentation of coal particle images. Coal samples exhibit complex particle morphology with significant variation between samples, making traditional segmentation methods challenging. SAM excels in adapting to diverse image characteristics and effectively completing segmentation tasks without requiring extensive labeled data [24]. For coal particle images, SAM accurately identifies and separates particles of different sizes, generating binary mask images that reflect coal particle morphology. Figure 5 demonstrates SAM’s strong capability in coal particle image processing. Compared to traditional methods, SAM not only improves segmentation accuracy but also significantly enhances the model’s adaptability to complex and dynamically changing scenarios. Using SAM for coal particle image segmentation provides high-quality data support for subsequent analyses such as coal quality detection and particle size effect correction, advancing coal analysis technology.

#### 2.2.3. PLSR Model

We previously developed an ash content prediction model for coal with a median particle size of 0.2 mm, implemented on our experimental platform using Partial Least Squares Regression (PLSR). We collected near-infrared and X-ray fluorescence spectral data. These data are preprocessed to reduce noise and enhance quality. Principal Component Analysis (PCA) is then applied to extract relevant features from the processed spectra, focusing on the principal components associated with ash content, such as specific wavelengths in the NIRS spectrum or element concentrations measured by XRF. A PLSR model is constructed to establish the relationship between the extracted features and ash content. The model’s output is the predicted ash content, which has not been corrected for particle size effects. The model was validated on 0.2 mm coal powder samples, demonstrating excellent performance with a repeatability standard deviation (SD) of less than 0.2%, indicating high accuracy in ash content prediction. The next step will involve addressing particle size effects through a correction model, targeting coal with a median particle size of 1 mm.

### 2.3. Correction Model

#### 2.3.1. DeiT Model

The core of the particle size effect correction model lies in using particle size distribution data to adjust the ash content of coal samples, effectively reducing measurement errors caused by particle size differences and improving the accuracy of ash measurement. The process begins by calculating the ash error as the difference between the actual ash content and the measured value of the sample. This error is then normalized and regularized, serving as the target value for ash error correction. For model design, 224 × 224 pixel three-channel color images are used as training inputs. These mask images incorporate comprehensive physical information about larger particles on the coal sample surface, including shape, color, position, and size.

In recent years, deep learning has achieved significant breakthroughs in image analysis [40], leading to the development of numerous efficient models. Among these, transformer models have demonstrated exceptional performance in tasks such as image classification, owing to their powerful feature extraction and global information modeling capabilities, exemplified by ViT [41]. The core mechanism of transformer models is self-attention, which captures long-range dependencies within images and provides a global receptive field. In contrast, CNN models [42] have local receptive fields that require multiple layers of convolution and pooling to gradually expand their receptive field. However, traditional transformer models require large datasets for training, which poses a challenge in coal composition analysis due to the high cost of acquiring large-scale coal image datasets.

To address this limitation, this study adopts the concept of transfer learning and employs the Data-efficient Image Transformers (DeiTs) Model. DeiT is an improved transformer model that uses a training method called “distillation” to enhance performance on small datasets. Distillation involves a knowledgeable teacher model (usually a high-performing traditional model, such as a CNN) guiding a student model (DeiT) to learn more effectively, as illustrated in Figure 6 [43]. This approach enables DeiT to learn meaningful image features from smaller datasets, which is crucial for analyzing coal images with limited data.

Specifically, the task involves mapping coal images to a floating-point number representing the ash error (actual ash value minus the measured value), essentially performing image regression. The DeiT model’s core mechanism, self-attention, efficiently captures relationships between different regions of an image, which is critical for understanding the microstructure and compositional distribution of coal. By pretraining on large general-purpose image datasets, DeiT learns generic image features. Subsequently, the pretrained DeiT model is fine-tuned on the coal image dataset to better adapt to coal-specific characteristics and accurately predict relevant physical and chemical properties. The structural design and workflow based on the DeiT deep learning model are as follows:Backbone Model: This study uses DeiT Base as the backbone model. Its architecture is based on Vision Transformer (ViT) and consists of multiple transformer encoders capable of extracting global features from input images. The model accepts 224 × 224 × 3 images as input, performs linear projection and positional encoding, divides the image into 16 × 16 patches, and maps these patches into fixed-length embedding vectors. Each embedding vector is processed through a multi-head self-attention mechanism and a feed-forward network to capture global feature relationships.Adaptation for Regression Tasks: To adapt to the regression task for ash correction, the DeiT classification head is replaced with a regression head. The original classification head is modified to a linear layer with a single output node for predicting continuous values. The input dimension of the new linear layer matches the final embedding features of DeiT (768 dimensions), while the output dimension is set to 1 to generate ash correction predictions.Feature Extraction and Regression Output: Preprocessed images are fed into the backbone DeiT model, where multi-layer transformer encoders extract global image features. The regression head then produces the ash correction prediction. The transformer structure effectively models complex spatial distributions and ash variation patterns within coal particle images through its global modeling capability.

Compared to traditional CNNs, DeiT directly models global feature interactions through its multi-head self-attention mechanism, eliminating the dependency of convolution operations on local regions. Furthermore, DeiT’s data efficiency allows it to perform well on smaller labeled datasets by leveraging pretrained models. By adapting the regression head to task-specific needs, this method achieves a direct mapping from coal particle images to ash predictions, providing robust support for multimodal feature integration and analysis of complex feature relationships.

Unlike traditional image analysis methods that rely on handcrafted features, DeiT automatically learns intricate image characteristics, avoiding the complexity and subjectivity of manual feature design. Its global receptive field further enhances its ability to capture the overall structure of coal images, which is crucial for composition analysis tasks requiring consideration of global structures. As a complementary tool to CNNs, DeiT provides a powerful modeling approach for ash correction tasks, improving both prediction accuracy and generalization capability. Therefore, using DeiT for regression tasks to correct ash measurement errors is highly appropriate.

#### 2.3.2. Model Evaluation

To evaluate the established particle size effect correction model, this study uses Standard Deviation (SD) and Root Mean Square Error (RMSE) as assessment metrics to measure the model’s accuracy and repeatability in prediction results.

SD measures the model’s repeatability, reflecting the consistency of ash prediction results obtained from multiple measurements of the same coal sample under identical conditions. A smaller SD value indicates better repeatability of the model’s ash predictions. The calculation formula is as follows:(1)$$SD=\sqrt{\frac{\sum_{i=1}^{n}{\left(X_{i}-\bar{X}\right)}^{2}}{n-1}}$$
where $X_{i}$ represents the corrected ash value from the i-th measurement of the same coal sample, $\bar{X}$ represents the average of all predicted values for the coal sample during repeated measurements, and n denotes the total number of repeated measurements.

RMSE is a widely used error evaluation metric primarily designed to quantify the deviation between the model’s predicted ash values and the actual ash values of coal samples. Compared to Mean Absolute Error (MAE), RMSE is more sensitive to larger errors due to its inclusion of a squared term, which amplifies the impact of significant errors. Therefore, RMSE is better suited for scenarios requiring a focus on penalizing large errors and more effectively reflects the model’s stability and precision in predictions. The calculation formula is as follows:(2)$$RMSE=\sqrt{\frac{\sum_{i=1}^{n}{\left(Y_{i}-\bar{Y}\right)}^{2}}{n}}$$
where $Y_{i}\;$ represents the actual ash value of the coal sample, $\bar{Y}\;$ represents the predicted value, and *n* denotes the total number of samples.

## 3. Results

### 3.1. Impact of Particle Size on NIRS and XRF Spectra

In the experimental analysis, the reproducibility of NIRS spectra (Figure 7) and XRF energy spectra (Figure 8) was compared for the same coal sample under different particle size conditions (median diameters of 0.2 mm and 1 mm). The coal sample surface was leveled multiple times, and five repeated measurements were conducted under each particle size condition.

The results show that changes in surface particle distribution significantly affected spectral stability, with larger-particle samples exhibiting greater spectral fluctuations compared to smaller-particle samples. Spectral stability refers to the consistency of spectral data over time and under varying conditions, characterized by minimal spectral fluctuations. In our experiment, smaller-particle samples maintained more consistent spectral characteristics after each leveling, indicating better spectral stability.

For NIRS spectra, 1 mm particle samples displayed higher absorbance. Larger particles caused stronger light scattering and longer optical path lengths, enhancing light–sample interaction and increasing absorbance. Poor surface uniformity in large-particle samples further increased light capture and absorption efficiency due to variations in porosity. Additionally, the uneven distribution of chemical components accentuated absorption peaks at specific wavelengths.

For XRF energy spectra, characteristic emission lines of various elements were labeled. Small-particle samples exhibited significantly higher fluorescence intensity for elements such as S, Ti, Fe, and Co. This is because fine-particle samples had a more uniform surface, enhancing fluorescence signal output, while larger particles with rough surfaces and larger particle spacing caused signal attenuation.

### 3.2. Training and Evaluation of the Correction Model

During the training and evaluation phase of the ash correction model, we used the SAM model to extract particle size features in the task of precise coal particle segmentation in microscopic images, ensuring the accuracy of the particle size effect correction model training.

To adapt to this specific task, we adjusted the SamAutomaticMaskGenerator parameters in the SAM model to minimize the impact of large coal particles on the segmentation results. The optimized parameters are detailed in Table 1. These adjustments significantly reduced the interference caused by large coal particles, making the segmentation results more aligned with the true morphology of the coal particles.

Figure 9 illustrates the complete workflow from capturing the raw images to generating the final segmented coal particle images. Initially, the raw images are acquired at a resolution of 1920 × 1080, containing mixed information of coal particles and background. To improve the effectiveness of subsequent segmentation, the images undergo preprocessing, including cropping and denoising, resulting in images resized to 1000 × 1000 pixels.

The preprocessed images are then input into the SAM model for segmentation, which generates multiple binary segmentation instances, each representing an independent coal particle region. During segmentation, the instances output by the SAM model are visualized using randomly assigned colors with a transparency of 0.6 to facilitate evaluation and verification of the segmentation results. This produces an integrated mask preview.

Next, the segmentation results are optimized through filtering and overlay operations, retaining only regions that meet the particle size requirements. This step generates the final binary segmentation mask (a single-channel image) that clearly delineates coal particle regions while masking irrelevant background information.

The binary mask is then used to extract coal particle regions from the original images. The extracted images are standardized by resizing them to 224 × 224 × 3 pixels, making them suitable as input for deep learning models. This comprehensive workflow, combining image preprocessing, segmentation, filtering, and standardization, ensures the generated images are of high quality and consistency, providing reliable input data for subsequent analysis and modeling.

This processing pipeline demonstrates the efficiency of the SAM model, not only achieving precise coal particle segmentation but also eliminating over-segmentation and regions with abnormal sizes through post-processing. This significantly improves the consistency between the segmentation mask and the actual geometry of coal particles, providing high-quality input data that enhance the accuracy and stability of the ash prediction model.

Figure 10 shows five coal particle images from the dataset. The top row represents the original color images with circular masks applied, while the bottom row displays the color images generated after segmentation and post-processing by the SAM model. The results demonstrate the SAM model’s robust ability to extract and segment individual coal particles from images, producing outputs closely aligned with the actual geometric features of coal particles.

During the training phase, a custom regression model based on the DeiT architecture was implemented using the PyTorch [44] framework and executed on a GPU to accelerate computation. The MSELoss function was selected as the loss criterion, and the Adam optimizer was used with an initial learning rate of 0.0001, ensuring stable training and convergence.

To further enhance the model’s training effectiveness and adaptability, a Cosine Annealing Learning Rate Scheduler was introduced. This scheduler dynamically adjusts the learning rate over the total number of training epochs (num_epochs), gradually decaying it from the initial value to a minimum value of 0.00001. This learning rate strategy allows the model to explore a broader learning space in the early stages of training and refine its parameters during later stages through the gradual reduction in the learning rate.

## 4. Discussion

### 4.1. Significance of Particle Size Effects on Spectral Stability

The results highlight the critical impact of particle size on the stability of NIRS and XRF spectra. Large-particle samples exhibited greater spectral fluctuations due to factors such as stronger light scattering, uneven porosity, and inconsistent chemical composition. For NIRS, higher absorbance in larger particles was attributed to enhanced light-sample interaction. For XRF, the fluorescence intensity was significantly higher in fine-particle samples due to their uniform surface. These findings underscore the necessity of correcting for particle size effects to improve measurement accuracy and stability.

### 4.2. Impact of Correction on Ash Prediction Accuracy

The validation set experiments showed significant improvements in the SD of ash predictions after correction (Figure 11). For example, the SD of sample #1 decreased from 0.37% to 0.30%, and sample #2 from 0.35% to 0.28%. Similarly, the SD of samples #3 and #4 dropped from 0.33% and 0.32% to 0.26% and 0.24%, respectively. Other samples, such as #5, #6, #7, and #8, also exhibited varying degrees of reduction, with sample #6 showing the most notable improvement, decreasing from 0.34% to 0.27%. Overall, the average SD of ash predictions decreased from 0.34% to 0.27% after correction, representing a reduction of approximately 20.59%. These results indicate that particle size effect correction not only improves the accuracy of ash predictions but also enhances their repeatability, providing more reliable support for coal quality analysis.

Figure 12 presents the changes in RMSE of ash predictions before and after correction. After correction, RMSE showed a significant reduction. For instance, the RMSE of sample #1 decreased from 0.40% to 0.34%, sample #3 from 0.37% to 0.30%, sample #4 from 0.36% to 0.26%, and sample #8 from 0.39% to 0.27%. Overall, the average RMSE of the samples decreased from 0.36% to 0.28% after correction, representing a reduction of approximately 22.22%. These results clearly demonstrate that particle size effect correction significantly improves the model’s predictive accuracy and adaptability to samples with different particle sizes.

In summary, the model correction significantly improved the performance of ash prediction in terms of both SD and RMSE: the reduction in SD indicates enhanced stability and repeatability of the model’s predictions, while the decrease in RMSE reflects reduced prediction errors, improved accuracy, and more consistent predictive performance across different samples.

## 5. Conclusions

To address the impact of particle size effects on the accuracy of spectral analysis for coal samples, this study proposed a correction method based on the SAM and DeiT models. This method successfully established a more comprehensive and efficient particle size effect correction mechanism, offering a novel technical solution to tackle complex interference factors in coal quality analysis.

During the experiments, the DeiT model was utilized to extract deep features from microscopic images of coal samples. Leveraging DeiT’s transformer architecture, the model captured comprehensive global information on particle size distribution and morphological characteristics. Subsequently, DeiT’s regression capability modeled the extracted high-dimensional features, learning the complex nonlinear relationships between particle size features and target parameters, such as ash content.

Compared to traditional spectral correction methods, this approach eliminates the reliance on spectral data alone by integrating spatial distribution characteristics and geometric structures of coal samples into the correction model. This significantly enhances adaptability and correction accuracy for particle size effects.

The experimental results demonstrate that the DeiT-based correction method substantially improved ash prediction accuracy, reducing SD from 0.34% to 0.27% and RMSE from 0.36% to 0.28%. These outcomes highlight the method’s significant advantages in mitigating particle size effect interference and enhancing the precision of spectral analysis.

The innovation of this study lies not only in the application of the DeiT model but also in the in-depth exploration and utilization of complex spatial characteristics of samples. By incorporating the global modeling capabilities of transformer models, this method constructs a precise and generalizable particle size effect correction mechanism, providing a novel solution for handling complex interferences in coal quality analysis. Moreover, the method has broad application potential, extending to other tasks requiring particle size effect correction in complex sample analyses, and laying a foundation for the innovative development of spectral analysis technology.

The particle size effect correction method proposed in this study has potential applications not only in coal quality analysis but also in various other industries. For instance, in environmental monitoring, this method may improve the accuracy of pollutant concentration predictions in soil and water samples. In the mining sector, it could enhance ore composition testing and potentially improve assessment precision. Furthermore, in the field of food science, this method might aid in better particle analysis of powdered food products, contributing to a deeper understanding of their compositional characteristics. In medical diagnostics, particularly in cancer detection, this approach may help analyze the particle characteristics of biological samples, potentially improving disease prediction. Additionally, drug development may benefit from this method by optimizing the analysis of drug release characteristics and possibly increasing the accuracy of drug efficacy assessments. In summary, this research offers a novel perspective on addressing particle size effects in the analysis of various complex samples, and we will continue to explore its broader application prospects in the future.

## Figures and Tables

**Figure 1 sensors-25-00928-f001:**
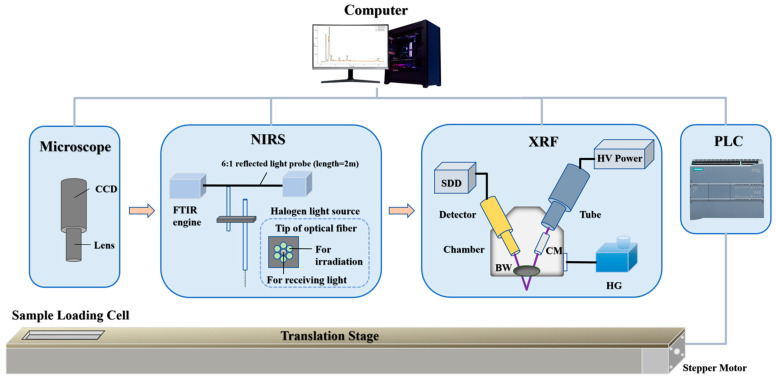
NIRS-XRF combined coal quality analysis setup for particle size effect correction (CCD: charge-coupled device, FTIR: Fourier-transform infrared spectroscopy, HV Power: high voltage power, HG: hydrogen generator, BW: beryllium window, CM: collimator, SDD: silicon drift detector, PLC: programmable logic controller).

**Figure 2 sensors-25-00928-f002:**
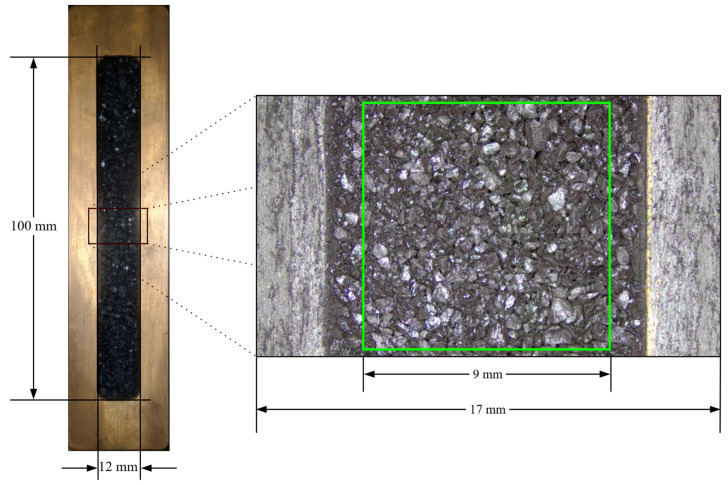
Schematic diagram of the sample cell and the corresponding magnified image showing the measurement area.

**Figure 3 sensors-25-00928-f003:**
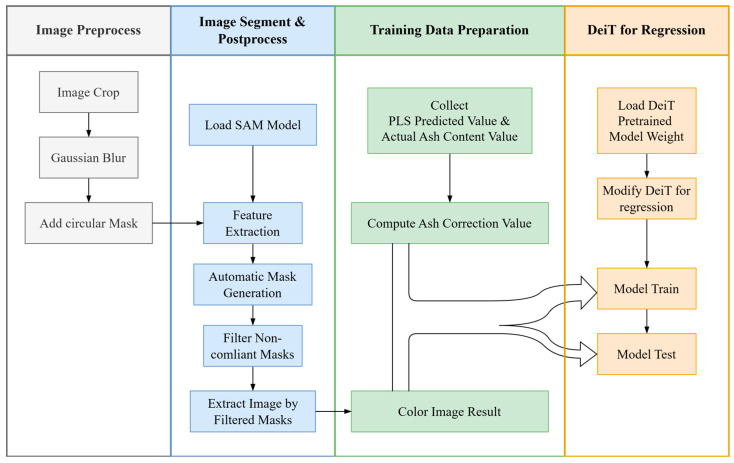
Overall construction process of the particle size effect correction model.

**Figure 4 sensors-25-00928-f004:**
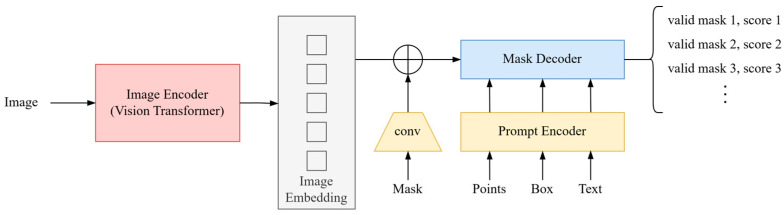
Basic structure of the SAM model.

**Figure 5 sensors-25-00928-f005:**
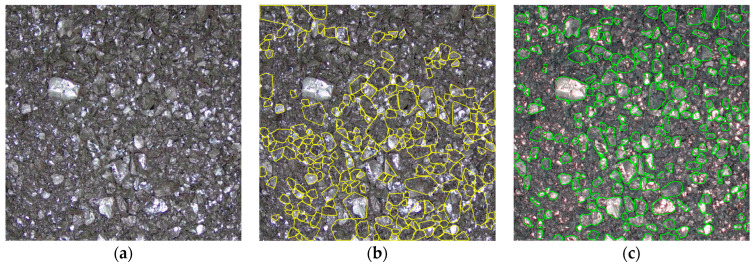
Comparison of different segmentation methods. (**a**) Coal sample original microscopic image; (**b**) Watershed segmentation using convex hull analysis; (**c**) SAM segmentation.

**Figure 6 sensors-25-00928-f006:**
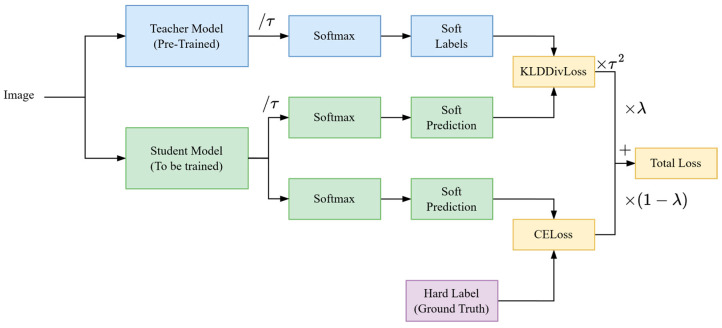
Teacher–student distillation training in DeiT model.

**Figure 7 sensors-25-00928-f007:**
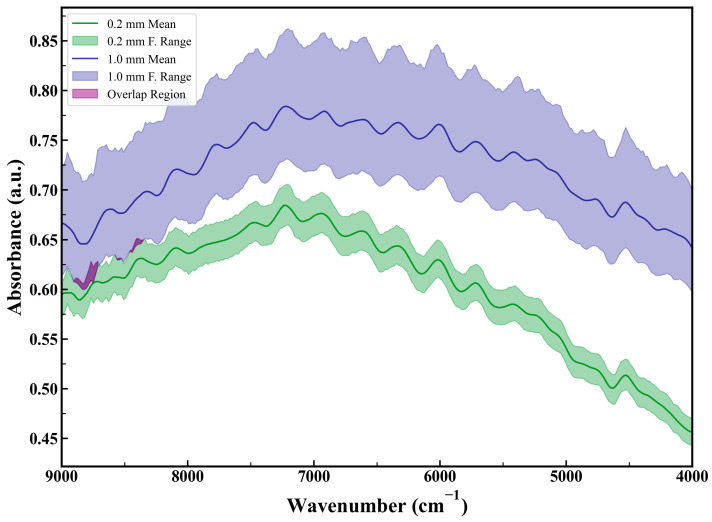
NIRS spectra of the same coal sample leveled repeatedly with different particle sizes.

**Figure 8 sensors-25-00928-f008:**
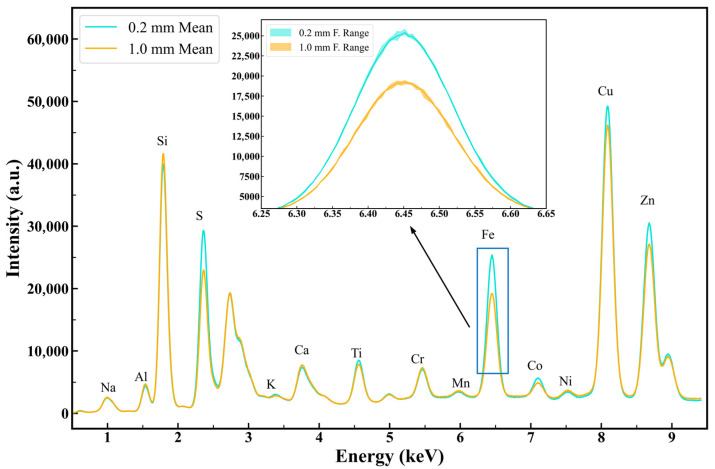
XRF energy spectra of the same coal sample leveled repeatedly with different particle sizes.

**Figure 9 sensors-25-00928-f009:**
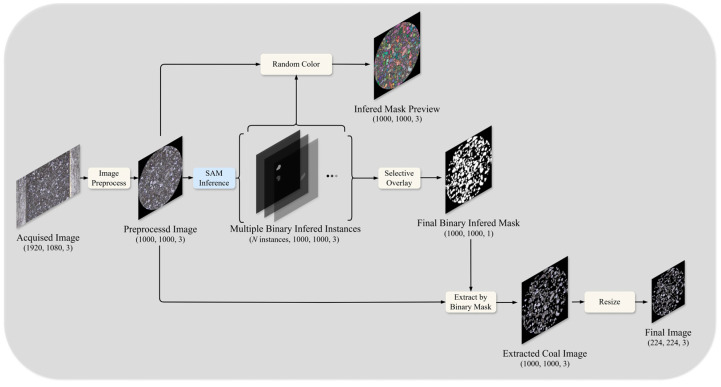
Image processing workflow.

**Figure 10 sensors-25-00928-f010:**
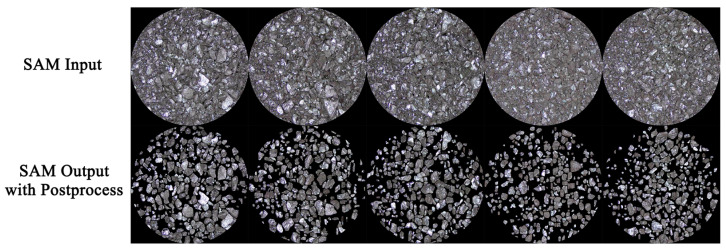
Comparison of coal particle images in the dataset.

**Figure 11 sensors-25-00928-f011:**
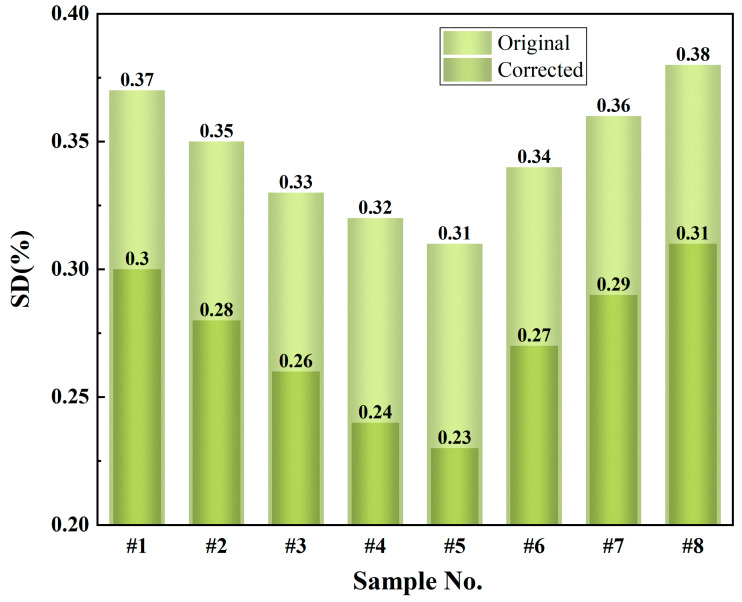
Comparison of standard deviation (SD) before and after correction.

**Figure 12 sensors-25-00928-f012:**
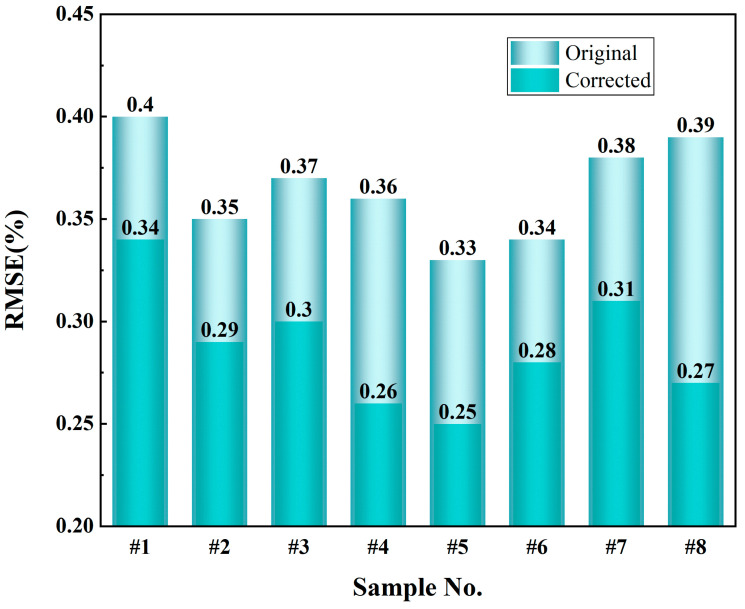
Comparison of root mean square error (RMSE) before and after correction.

**Table 1 sensors-25-00928-t001:** Parameter settings for the SamAutomaticMaskGenerator.

Parameter	Value	Type
points_per_side ^1^	32	int
points_per_batch ^2^	128	Int
pred_iou_thresh ^3^	0.87	float
stability_score_thresh ^4^	0.8	float
box_nms_thresh ^5^	0.8	float
min_mask_region_area ^6^	200	int

^1^ The number of points to be sampled along one side of the image. ^2^ Sets the number of points run simultaneously by the model. ^3^ A filtering threshold in [0,1], using the model’s predicted mask quality. ^4^ A filtering threshold in [0,1], based on the mask’s stability when changing the cutoff for binarizing the mask. ^5^ The box IoU cutoff used by non-maximal suppression to filter duplicate masks. ^6^ Remove disconnected regions and holes in masks with area smaller than min_mask_region_area.

## Data Availability

The raw data supporting the conclusions of this article will be made available by the authors on request.

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
