# Peer review of "DeiT and Image Deep Learning-Driven Correction of Particle Size Effect: A Novel Approach to Improving NIRS-XRF Coal Quality Analysis Accuracy"

_sensors, 2025, doi:10.3390/s25030928_

Round 1

Reviewer 1 Report

Comments and Suggestions for Authors

The manuscript presents an innovative correction method that integrates the Segment Anything Model (SAM) for precise particle segmentation and Data-efficient image Transformers (DeiT) for ash prediction modeling to address the particle size effect in NIRS-XRF coal quality analysis. This research tackles a significant challenge in coal analysis and proposes a solution that effectively bridges traditional spectroscopy with advanced machine learning techniques. The paper is well-organized, and the experimental results demonstrate notable improvements in prediction accuracy and consistency. The study's novelty lies in its integration of SAM and DeiT, which represents a notable contribution to the field. The methodology is robust, combining advanced image segmentation with regression modeling to achieve practical results. Furthermore, the research addresses a critical issue in coal quality analysis with strong real-world applicability. The significant reduction in SD and RMSE, supported by experimental data, highlights the effectiveness of the proposed approach. The manuscript demonstrates significant contributions to the field and aligns well with the scope of Sensors. I recommend accepting the manuscript with minor revisions, as detailed below:

1. The legend of the diagram is unclear. For example, what does CM stand for in Figure 1? It looks like a collimator but there is no explanation in the legend. Conversely, the HV Power in the legend is not reflected in the figure.

2. The manuscript briefly lists the parameters for the Sam Automatic Mask Generator (Table 1). It would be helpful to provide a rationale or discussion for the chosen parameter values. Additionally, it would be valuable to clarify whether the parameter selection was optimized using specific methods or experiments.

3. While the study discusses potential broader applications of the proposed method, a short paragraph summarizing specific industries or scenarios (beyond coal analysis) would add value.

Comments on the Quality of English Language

no

Author Response

Comments 1: The legend of the diagram is unclear. For example, what does CM stand for in Figure 1? It looks like a collimator, but there is no explanation in the legend. Conversely, the "HV Power" mentioned in the legend is not reflected in the figure.

Response 1: Thank you for highlighting this issue. We agree with your comment and have revised the figure legends accordingly. In Figure 1 on page 3 of the manuscript, we have added detailed explanations for the terms CCD, Power, CM, and PLC in the legend to ensure greater clarity for readers. These revisions aim to eliminate ambiguities and align the legend with the visual elements in the figure. We appreciate your feedback and believe these updates enhance the overall understanding of the figure.

Comments 2: The manuscript briefly lists the parameters for the Sam Automatic Mask Generator (Table 1). It would be helpful to provide a rationale or discussion for the chosen parameter values. Additionally, it would be valuable to clarify whether the parameter selection was optimized using specific methods or experiments.

Response 2:

Thank you for your insightful suggestion. We have expanded the explanation of the parameter selection process and included a detailed rationale for the chosen values in Table 1. On page 11 of the manuscript, we now provide a comprehensive discussion of the optimization methods and experimental processes used. Below is a detailed description of the parameters:

  • Points_per_side (32): This parameter defines the number of points sampled along one side of the image. Based on preliminary experiments, a value of 32 was chosen as it balances computational efficiency with sufficient coverage of image details, enabling the model to effectively capture coal particle boundaries.
  • Points_per_batch (128): This determines the number of points processed simultaneously. A batch size of 128 was selected to optimize computational performance, ensuring sufficient information is processed per iteration without exceeding memory constraints during training or inference.
  • Pred_iou_thresh (0.87): The IoU threshold for filtering predicted masks was set to 0.87 to retain high-quality masks while discarding less accurate predictions. This value was fine-tuned through experimental testing, with higher thresholds yielding better segmentation consistency, particularly for larger particles.
  • Stability_score_thresh (0.8): This threshold controls the stability of mask predictions when binarization cutoffs are adjusted. A value of 0.8 was found to provide an optimal tradeoff between mask stability and sensitivity, especially for capturing irregular coal particle shapes.
  • Box_nms_thresh (0.8): Non-maximum suppression (NMS) was applied with an IoU threshold of 0.8 to filter overlapping mask proposals. This value minimizes duplicate detections without discarding true-positive masks, particularly in densely packed particle areas.
  • Min_mask_region_area (200): A minimum mask region area of 200 was used to eliminate noise and irrelevant small regions. This value was determined based on the average size of coal particles in the dataset, ensuring only meaningful regions are retained.

The parameter values were optimized through iterative experimentation using a representative subset of the dataset. During this process, we systematically adjusted and refined each parameter to identify the optimal configuration that achieved the best segmentation performance. Evaluation criteria included quantitative metrics such as Intersection over Union (IoU) and qualitative visual assessments. By analyzing segmentation results across multiple trials, we ensured that the chosen parameters consistently aligned with the true morphology of coal particles. This iterative approach accounted for variations in particle size and shape, improving model robustness and segmentation accuracy. We believe this detailed explanation provides greater clarity and value to readers.

Comments3: While the study discusses potential broader applications of the proposed method, a short paragraph summarizing specific industries or scenarios (beyond coal analysis) would add value.

Response3:

Thank you for this valuable suggestion. We have incorporated a new paragraph in Section 5 of the manuscript to highlight the potential broader applications of the proposed method. Below is the added content:

The particle size effect correction method proposed in this study holds significant potential beyond coal quality analysis. For instance, in environmental monitoring, this approach could improve the accuracy of pollutant concentration predictions in soil and water samples. In the mining industry, it may enhance ore composition testing, leading to improved precision in resource assessments. Furthermore, in the field of food science, this method could aid in the analysis of powdered food products, providing insights into their compositional characteristics. In medical diagnostics, particularly in cancer detection, this method may assist in analyzing the particle characteristics of biological samples, potentially enhancing disease prediction. Additionally, in drug development, this technique could optimize the analysis of drug release profiles, increasing the accuracy of efficacy assessments. Overall, this research provides a novel perspective on addressing particle size effects across various complex samples, and we aim to further explore its broader applications in future work.

Reviewer 2 Report

Comments and Suggestions for Authors

The article is devoted to the use of machine learning for segmentation of microphotos of coal samples, which are then used to correct the results of IR and X-ray fluorescence spectroscopy. The authors proposed an original data processing scheme, as well as a setup for experimental verification of the processing method.

The article makes a good impression and will be of interest to readers of the journal.

As minor comments, I would like to note the following.

It is advisable for the authors to describe the experimental setup in more detail. In particular, it is necessary to provide a diagram from which it will be clear which fragment of the sample surface is recorded as a microphoto, which fragment is used to capture IR and X-ray spectra. It is necessary to provide the sizes of the analyzed areas.

The term "spectra stability" should be clarified.

It is advisable to study the size distribution of coal particles using an independent method (e.g. dynamic light scattering).

It is necessary to describe how ash content is calculated directly from IR and XRF data, without using correction models.

In the opinion of the reviewer, the article can be published after minor revision.

Author Response

Comments 1: It is advisable for the authors to describe the experimental setup in more detail. In particular, it is necessary to provide a diagram from which it will be clear which fragment of the sample surface is recorded as a microphoto, which fragment is used to capture IR and X-ray spectra. It is necessary to provide the sizes of the analyzed areas.

Response 1: Thank you for your valuable comment. We agree with your suggestion and have added more detail regarding the experimental setup. Specifically, we have included a diagram (Figure 2 in new uploaded manuscript) of the sample loading cell in section 2.1.1, clearly labeled with the actual sizes of the analyzed areas. In the experiment, it is crucial that the cropped region of the microscopic images precisely aligns with the spots for IR and X-ray spectra to ensure accurate multispectral data fusion and effective calibration. Therefore, the microphoto, IR, and X-ray spectra were all collected from the green square area depicted in Figure 2.

Comments2: The term "spectral stability" should be clarified.

Response 2:

Thank you for pointing out the need for clarification of the term "spectral stability." We have added a detailed explanation in Section 3.1.

Spectral stability refers to the consistency of spectral data over time and under varying conditions, with minimal fluctuations in intensity and shape across repeated measurements or different sample preparations. It is essential for reliable and reproducible analysis, ensuring that external factors like particle size, surface roughness, or environmental conditions do not cause significant variation in the spectral data.

In our experiment, spectral stability was assessed by evaluating the fluctuation of spectral data after each leveling process. We observed that finer particles exhibited greater spectral consistency, as they caused fewer surface irregularities and more uniform spectra. In contrast, larger particles led to greater fluctuations due to uneven particle distribution on the sample surface.

Comments3: It is advisable to study the size distribution of coal particles using an independent method (e.g. dynamic light scattering).

Response3:

Thank you for your thoughtful suggestion. We appreciate the recommendation to study the size distribution of coal particles using an independent method, such as dynamic light scattering. However, due to the upper size limit of the coal particles being in the millimeter range, there are limited methods available for precise particle size measurement at this scale. For example, laser diffraction, with a measuring range of 0.01 µm to 4 mm, is a method well-suited for coal particle analysis. While laser diffraction provides rapid and effective measurement of particle size distribution, it has limitations in recognizing particle shape and irregularities. Laser diffraction primarily focuses on particle size but cannot capture detailed shape and surface characteristics.

In future work, we plan to apply laser diffraction, combined with image analysis, to conduct independent particle size distribution analyses. The image analysis can provide additional shape and surface texture information, complementing the laser diffraction results and improving the overall accuracy. This will allow us to validate the effectiveness of the image-based particle size analysis method we currently use.

Comments 4: It is necessary to describe how ash content is calculated directly from IR and XRF data, without using correction models.

Response 4:

Thank you for your valuable comment. We agree with this suggestion and have clarified the process of directly calculating ash content from NIRS and XRF data without relying on correction models. This process is detailed in Section 2.2.3 of the manuscript, where we describe the Partial Least Squares Regression (PLSR) model developed on our experimental platform for coal with a median particle size of 0.2 mm.

First, near-infrared spectral (NIRS) data and X-ray fluorescence (XRF) data are collected from the samples. These raw data are preprocessed to reduce noise and enhance their quality. Principal Component Analysis (PCA) is then applied to extract key features from the processed spectra, focusing on principal components associated with ash content, such as specific wavelengths in the NIRS spectrum or element concentrations measured by XRF. A PLSR model is subsequently constructed to establish a mathematical relationship between the extracted features and ash content, enabling the direct calculation of ash content based on the intensity of these feature bands or element concentrations without using the correction model.